# Application of Bivariate Reproducing Kernel-Based Best Interpolation Method in Electrical Tomography

**DOI:** 10.3390/s24227165

**Published:** 2024-11-07

**Authors:** Yongguang Tan, Jingqi Wang, Junqi Yu, Boqi Wu, Jinchao Shen, Xiangchen Guo

**Affiliations:** 1School of Building Services Science and Engineering, Xi’an University of Architecture and Technology, Xi’an 710055, China; tanyongguang2023@xauat.edu.cn (Y.T.); wjqmo@xauat.edu.cn (J.W.); shenjinchao@xauat.edu.cn (J.S.); o2316212720@xauat.edu.cn (X.G.); 2Key Laboratory of Architectural Cold Climate Energy Management, Jilin Jianzhu University, Ministry of Education, Changchun 132600, China; wuboqi@jlju.edu.cn

**Keywords:** electrical tomography, interpolation approximation, bivariate reproducing kernel, spatial resolution

## Abstract

Electrical Tomography (ET) technology is widely used in multiphase flow detection due to its advantages of low cost, visualization, fast response, non-radiation, and non-invasiveness. However, ill-posed solutions lead to low image reconstruction resolution, which limits its practical engineering applications. Although existing interpolation approximation algorithms can alleviate the effects of the ill-posed solutions to some extent, the imaging results remain suboptimal due to the limited approximation capability of these methods. This paper proposes a Bivariate Reproducing Kernel-Based Best Interpolation (BRKBI) method, which offers smaller approximation errors and clearer image reconstruction quality compared to existing methods. The effectiveness of the BRKBI method is validated through theoretical analysis and experimental comparisons.

## 1. Introduction

Since its emergence in the 1980s, Electrical Tomography (ET), an advanced, radiation-free, non-invasive, and cost-effective detection technology, has attracted significant attention and has been extensively researched across various fields, including industrial inspection [1], medical diagnostics [2], and transportation [3]. However, due to ill-posed solutions [4] and the soft-field effect [5], ET technology faces limitations such as low spatial resolution in visualized imaging and poor object image reconstruction capabilities, particularly in detecting complex structures. This paper primarily addresses ill-posed solutions, which in electrical tomography refers specifically to the situation where the available measurements are far fewer than the number of pixels in the reconstructed image, resulting in low spatial resolution. To overcome this issue, over the past decade, researchers have conducted extensive studies on both hardware and software, proposing numerous new methods and achieving a series of advancements.

Research on hardware has primarily focused on ET (Electrical Tomography) measurement systems, which can be broadly categorized into the following four areas: increasing the number of electrodes, modifying measurement modes, expanding the range of excitation frequencies, and innovating excitation methods.

In the research on increasing the number of electrodes, researchers have expanded independent measurements by adding more electrodes, with the rotating ET system being a notable example [6], which enhances the number of independent measurements by moving the electrodes from their original positions to subsequent locations. This approach helps improve the resolution of ET images while mitigating the image quality issues caused by ill-posed solutions, thereby resulting in better overall ET image quality.

In the research on altering measurement modes, researchers have aimed to improve the imaging quality of reconstructed images by obtaining additional measurements through multiple ET modes under different excitations. For example, more measurements can be acquired using eight typical measurement modes, and the interactions between these ET modes can be represented using an integration method [7]. Subsequently, the optimal measurement combination is selected from the eight ET modes to reconstruct the ET image.

In the research on the excitation frequency range, researchers have leveraged the complementarity between measurements at various frequencies in ET image reconstruction [8]. By employing different frequency pairs for excitation, they have obtained additional measurements, thereby enhancing the quality of ET images. Although significant progress has been made through these aforementioned methods, the development of ET hardware systems has now reached a bottleneck.

In terms of software algorithms, the existing ET imaging algorithms primarily include linearized algorithms and nonlinear iterative algorithms [9]. Nonlinear iterative algorithms, which typically achieve one-step imaging, are particularly well-suited for real-time detection and control in engineering applications. Representative algorithms include classic methods such as the Linear Back Projection (LBP) algorithm [10], the sensitivity matrix algorithm [11], the pre-iterative method [12], the one-step Tikhonov Regularization method [13]. These classic methods have unique characteristics and applicable scenarios, yet they also face issues like artifacts and low resolution. To enhance the reconstruction quality of these algorithms, many researchers have made advancements based on them in recent years. For example, Wang et al. applied Density Peaks Clustering (DPC) and the K-means method to the LBP algorithm, significantly improving the image reconstruction effect [14]. Zhang et al. proposed a nonlinear, non-convex image reconstruction algorithm based on the homotopy method, transforming the ERT inverse problem into a multi-objective non-convex optimization that enhances reconstruction accuracy and avoids local optima [15]. Li et al. introduced an improved regularization reconstruction method by using homotopic mapping to select the regularization parameter in the iterative Tikhonov algorithm, thereby enhancing the image reconstruction accuracy [16].

Additionally, some researchers have improved reconstruction accuracy by integrating different reconstruction algorithm theories. For instance, Guo et al. proposed a new truncation strategy for the regularized D-bar method for ERT image reconstruction. By applying Tikhonov regularization theory to solve the scattering transform of the D-bar algorithm, they improved reconstruction accuracy [13]. With the rise in machine learning methods and their powerful computational analysis capabilities, many researchers have applied them to enhance ERT system image reconstruction quality. For example, Lan et al. developed a novel Monte Carlo (MC)-Net ML scheme that uses an MC dropout strategy with multiple stochastic methods to approximate Bayesian inference, allowing for quantitative estimation of uncertainty in ERT image reconstruction and thereby improving image reconstruction quality [17]. Yan et al. proposed a one-dimensional structure reparametrized convolutional neural network (1D-SRPCNN) algorithm for two-phase flow pattern image reconstruction based on ERT, which has been experimentally validated for high accuracy and efficiency in image reconstruction [18]. Similarly, Tang et al. developed a one-dimensional multi-branch convolutional neural network (1D-MBCNN) for ERT image reconstruction, incorporating an attention mechanism to improve reconstruction accuracy [19]. Liu et al. also proposed an ERT resolution enhancement network (ERTReNet) based on the popular U-Net architecture, improving the resolution of traditional inversion imaging results [20].

In addition to these methods, machine learning has also been applied to post-process reconstructed images. Huang et al. proposed an improved U-shaped deep learning model that combines the multi-scale feature extraction of UNet++ and the residual feature fusion of Res2Net to process pre-reconstruction results from traditional ERT methods, ultimately enhancing image quality [21]. Although these approaches aim to improve image reconstruction accuracy through various enhancements, the inherent underdetermination of these algorithms still significantly impacts reconstruction quality, often resulting in lower spatial resolution. To address this issue, some researchers have proposed new methods focusing on increasing the number of measurements. Notable examples include the Generalized Node Linear Back Projection (NLBP) algorithm [22]. and the Decomposing Linear Back Projection (DLBP) algorithm [23]. The NLBP algorithm reconstructs ET images by interpolating actual measurements to increase the number of measurements. The DLBP algorithm doubles the original number of measurements by decomposing the existing measurements. While both methods improve image resolution by mitigating ill-posed solutions to some extent, they also have certain limitations [24].

In recent years, researchers have proposed an optimal interpolation approximation algorithm [25] based on the theory of optimal interpolation approximation, known as the Reproducing Kernel-based Best Interpolation (RKBI) algorithm [26]. This method achieves optimal interpolation approximation for actual measurements within a Hilbert space equipped with reproducing kernels, thereby significantly increasing the number of applicable measurements. Compared to existing interpolation methods, the RKBI algorithm not only ensures optimal approximation but also provides estimable approximation errors. However, the RKBI algorithm uses a univariate reproducing kernel, which may result in unsatisfactory approximation capability and imaging performance under certain practical conditions. To enhance the algorithm’s performance, researchers both domestically and internationally have conducted in-depth studies over recent decades, achieving remarkable results in the theory and application of multivariate reproducing kernel hypothesis spaces. This research has overcome the limitations of approximation capability inherent in univariate reproducing kernel Hilbert spaces [27]. This paper introduces a Bivariate Reproducing Kernel-based Interpolation (BRKBI) method, which employs bivariate reproducing kernel functions to approximate measurements, thereby achieving smaller approximation errors compared to the RKBI method. The optimality of the BRKBI method is demonstrated through both theoretical analysis and experimental validation. Experiments show that BRKBI improves the accuracy and stability of ET images to some extent, providing a universal and innovative approach for enhancing spatial resolution in ET processes.

## 2. Related Work

Based on the different electrical parameters of the objects being detected in the measured electrically sensitive field, ET technology in practical applications is mainly categorized into three types: Electrical Resistance Tomography (ERT) [28,29], Electrical Capacitance Tomography (ECT) [30], and Electromagnetic Tomography (EMT) [31]. These correspond to quasi-static electric fields, static electric fields, and static magnetic fields, respectively. All three ET technologies adhere to the fundamental principles of electromagnetic fields, specifically Maxwell’s equations, and share the same mathematical mechanisms and physical laws. Therefore, in this paper, we use ERT as an example to illustrate our method.

This section is divided into two main parts. The first part introduces the principles of Electrical Resistance Tomography (ERT) measurement and several classical image reconstruction algorithms, discussing the advantages and shortcomings of existing interpolation approximation principles regarding image reconstruction quality in the context of the “underdetermined problem”. The second part provides a theoretical derivation to demonstrate the optimality of interpolation approximation in the bivariate reproducing kernel space, presenting the optimal interpolation function and the expression for approximation error. This serves as a theoretical foundation for the application of Bivariate Reproducing Kernel-based Interpolation in ERT systems in the third section.

### 2.1. ET Measurement Principles and Related Imaging Methods

The purpose of ET image reconstruction is to solve boundary value problems through boundary measurements to reconstruct the corresponding distributions of conductivity, dielectric constant, and magnetic permeability within the field. By mapping these distribution values for all pixels within the sensitive field to grayscale space, visualization of the target is achieved. We use a typical 16-electrode ERT system to illustrate the measurement and imaging principles.

As shown in Figure 1, an excitation voltage ‘V’ is applied to electrode 1 in the measurement field Ω, and measurements are taken from the remaining 15 electrodes, resulting in 15 measurement values. Next, the voltage ‘V’ is applied to electrode 2, and another set of 15 measurements is obtained. This process is repeated for all 16 electrodes, resulting in a total of 16 × 15 = 240 measurement values. These measurements are used to compute the conductivity distribution of the measurement field Ω to achieve image reconstruction.

Let σ denote the conductivity distribution function in the measurement field Ω. In fact, the relationship between the measurements σ and *u* at any point on *∂*Ω in the measurement field follows the Poisson equation [32], given by:(1)∇(σ ∇φ)=0,  s.t., u∈∂Ω

Here, *φ* represents the potential distribution within Ω, and *∂*Ω denotes the set of boundary measurements.

The essence of ERT image reconstruction is the solution of an inverse problem, which involves obtaining an approximate solution for the conductivity distribution σ within the measurement field by using a set of boundary measurements *u* from *∂*Ω.

By uniformly dividing the measurement field Ω into *m* units/pixels, each corresponding to a conductivity value δ_1_, δ_2_,…, δ_m_, the boundary measurements *u* can be expressed as *u*_1_, *u*_2_,…, *u*_n_. Based on the finite element method [33], the linearized discrete form of Equation (1) can be expressed as follows:(2)U=SG

Here, *G* = (δ_1_, δ_2_, …, δ_m_)^T^ and *U* = (*u*_1_, *u*_2_, …, *u*_n_)^T^; *S* represents the Jacobian matrix, also known as the sensitivity matrix [34], which depicts the sensitivity map from boundary excitations to the measurement units (pixels). The parameters *n* and *m* denote the number of boundary measurements on ∂Ω and the number of pixels within the measurement field Ω, respectively. Due to limitations in measurement hardware, it is typically the case that *m* > *n*. To solve for the variable *G* in Equation (2), different ERT algorithms are required depending on the specific application.

However, the ERT inverse problem is both nonlinear and ill-posed, which means that a direct analytical solution to Equation (2) does not exist. To address this issue, many methods have been proposed. Among these, the Linear Back Projection (LBP) algorithm is the most commonly used due to its highest time resolution. The solution form for the unknown *G* using the LBP algorithm is given as follows:(3)$$G=S^{T}U$$

However, because the LBP algorithm involves only a single iteration, its spatial resolution is very low.

The Tikhonov method, introduced in the 1960s, is an algorithm designed to address ill-posed problems and can be applied to the ET inverse problem. By incorporating prior information and regularization terms, the Tikhonov method reduces the solution space of the inverse problem, thereby stabilizing the solution matrix and making the problem well-posed.

In the ET inverse problem, the objective function for the Tikhonov regularization algorithm (minimization problem) is expressed as:(4)$$J(g)=\frac{1}{2}{\left\|Sg-z\right\|}^{2}+\mu R(g)$$

In this context, *R*(*g*) is the introduced regularization term, and *μ* is the regularization parameter. Typically, the regularization term can be expressed as:(5)$$R(g)={\left\|L(g-\bar{g})\right\|}^{2}$$

Here, *L* is the matrix that implements a specific operator, and $\bar{g}$ represents the estimated distribution of electrical characteristics (prior information). In the typical solution of the ET inverse problem, *L* is often replaced by the identity matrix, and $\bar{g}$ is replaced by a zero matrix. Therefore, Equation (4) becomes equivalent to solving:(6)$$(S^{T}S+\mu I)g=S^{T}z$$

In the equation, if the matrix *S^T^S* + *μI* is invertible, the solution to the above expression can be computed as:(7)$$\hat{g}={\left(S^{T}S+\mu I\right)}^{-1}S^{T}z$$

The regularization parameter *μ* is typically selected based on empirical experience and is usually a small positive value. The appropriate choice of *μ* significantly affects the quality of the image reconstruction performed by the regularization algorithm. The optimal regularization parameter *μ* can also be determined using methods such as the *L*-curve technique. Generally, the regularization parameter *μ* is sensitive, with different measurement models corresponding to different optimal values for *μ*. As observed from the above equation, the Tikhonov regularization algorithm also performs one-step imaging, which allows for fast imaging speeds and meets real-time requirements. In practical applications, the operator *L* in Equation (5) is often a diagonal matrix. Selecting *diag*(*S^T^S*) or *diag*(*sum*(*S^T^S*)) to replace the identity matrix can result in improved image reconstruction quality.

However, the ill-posed solution nature of ET measurements reduces the imaging quality of the above methods. To obtain more measurements and ensure that these measurements closely approximate the actual values, the RKBI algorithm has been proposed. This algorithm provides the optimal interpolation approximation for any measurement distribution function *u*(*x*) on the boundary ∂Ω of the ERT sensitive field Ω, with the approximation error being estimated. The principle is as follows: As shown in Figure 2, establish a polar coordinate system with electrode *T*_1_ as the origin to represent the 16-electrode ERT system. Apply excitation to electrode *T*_1_ and obtain 15 measurements *u*_1_, *u*_2_, …, *u*_15_ from the other electrodes. These 15 measurement electrodes correspond to polar coordinates (*ρ_j_*, *θ_j_*) in the polar coordinate system, where *j* = 1, 2,…, 15. According to Maxwell’s theory, the boundary measurements *u* are functions of *θ* and can be expressed as *u* = *F*(*θ*), with *θ* ∈ [0, 2*π*]. Since there are no actual electrodes at the interpolation points on the boundary, the measurement values cannot be directly obtained. To obtain the values of these interpolation points, it is necessary to use *u* = *F*(*θ*) and substitute the positions of these interpolation points expressed in terms of *θ* into the relationship to solve for *u*.

The RKBI algorithm is based on the theory of interpolation approximation in reproducing kernel Hilbert spaces. It derives the optimal interpolation function for the boundary measurements *u* and the boundary position parameters *θ*. Let {(*θ_j_*, *u_j_*)}_1_*^n^* be a set of given actual measurements, with *u_j_* = *F*(*θ_j_*). The expression is as follows:(8)$$F_{n}(\theta )=\sum_{j=1}^{n}a_{j}(\theta )u_{j},\;s.t.,\;a_{j}(\theta )=\int_{0}^{\theta }R_{\theta_{j}}^{⊥}(t)dt$$
where R*_θj_*^⊥^(*θ*) is a set of orthogonal basis functions obtained by applying Schmidt orthogonalization to the reproducing kernel R*_θj_*(*θ*).

By inserting *m* measurement points between the measurement electrodes, the values for 16*m* measurement points can be obtained using Equation (8). These 16*m* values are the optimal approximation of the 240 actual measurements. When combined with the original 240 actual measurements, the total number of measurements after interpolation and approximation becomes 240 + 16*m*. This effectively reduces the impact of the underdetermined problem on image reconstruction.

However, the interpolation approximation algorithm based on univariate reproducing kernels still faces several issues in practical image reconstruction, such as large approximation errors and suboptimal image reconstruction quality. In this paper, bivariate reproducing kernels will be used to perform interpolation approximation on the actual measurements to enhance the image reconstruction quality. The following sections will introduce how to construct bivariate reproducing kernels for interpolation approximation.

### 2.2. Interpolation Approximation Based on Bivariate Reproducing Kernels

The completeness, separability, and reproducing properties of multidimensional reproducing kernel spaces have been established [35]. We now construct a bivariate reproducing kernel space as follows:(9)$$W_{2}^{1}(\Omega^{2})=\left\{u(x,y)\left|u,\frac{\partial u}{\partial x},\frac{\partial u}{\partial y},\frac{\partial^{2}u}{\partial x\partial y}\in L^{2}(\Omega^{2})\right.\right\}$$

Let Ω^2^ = [a, b] × [a, b], where *a* and *b* are arbitrary real, and define the inner product ⟨·,·⟩_2_ and norm ∥·∥ in *W*_2_^1^. For any absolutely continuous functions *u*(*x*, *y*),*v*(*x*, *y*) ∈ *W*_2_^1^ (Ω^2^), let *M* = (*x*, *y*).
(10)$${\langle u(M),v(M)\rangle }_{2}=\int_{a}^{b}\int_{a}^{b}uvdxdy+\int_{a}^{b}\int_{a}^{b}\frac{\partial u}{\partial x}\frac{\partial v}{\partial y}dxdy+\int_{a}^{b}\int_{a}^{b}\frac{\partial u}{\partial y}\frac{\partial v}{\partial x}dxdy+\int_{a}^{b}\int_{a}^{b}\frac{\partial^{2}u}{\partial x\partial y}\frac{\partial^{2}v}{\partial x\partial y}dxdy$$
(11)$${\left\|u\right\|}_{2}\underline{\underline{def}}{\langle u,u\rangle }_{2}^{\frac{1}{2}}$$

Let *M* = (*x*, *y*) and *M_0_* = (*x*, *y*)′= (*ξ*, *η*). The reproducing kernel expression for this bivariate reproducing kernel space is defined as follows:(12)$$R_{M}(M_{0})=R_{\left(x,y\right)}(\xi ,\eta )≜R_{x}(\xi )R_{y}(\eta ),x,y,\xi ,\eta \in \left[a,b\right]$$
(13)$$R_{x}(\xi )=\frac{1}{2sh(b-a)}\left[ch(\xi +x-a-b)+ch(\left|\xi -x\right|+a-b)\right]$$
(14)$$R_{y}(\eta )=\frac{1}{2sh(b-a)}\left[ch(\eta +y-a-b)+ch(\left|\eta -y\right|+a-b)\right]$$

Here, *sh*(·) and *ch*(·) correspond to the hyperbolic sine and hyperbolic cosine functions, respectively.

Let *X* be a function space in *W*_2_^1^, and let {*M_i_* = (*x_i_*, *y_i_*)}_1_^2^ be a set of real numbers in the given interval Ω^2^. A set of functionals *I_i_*(*u*) is defined in space *X* as follows:(15)$$I_{i}(u)=u(M_{i})\underline{\underline{def}}\;u_{i}\;\;(u\in X,i=1,2,\cdots ,n)$$

Let *X_n_* be an *n*-dimensional subspace of *X*. Define the operator *H_n_* on *X* as follows:(16)$$H_{n}u(M)=\sum_{i=1}^{n}a_{i}(M)u_{i}\;\;\;(u(M)\in X,a_{i}(M)\in X_{n},i=1,2,\cdots ,n)$$
(17)$$E_{A}(X_{n};{\left\{a_{i}(M)\right\}}_{1}^{n})\equiv \underset{u\in A}{\sup}\left|\left(H_{n}u)(M)-u(M\right)\right|$$

Equation (17) represents the interpolation approximation error of *A* with respect to the interpolation basis set {*a_i_*(*M*)}_1_^2^ in *X_n_*.
(18)$$G_{A}(X_{n})\equiv \underset{\begin{array}{l} a_{i}(M)\in X_{n}\\ 1\leq i\leq n \end{array}}{\inf}E_{A}(X_{n};{\left\{a_{i}(M)\right\}}_{1}^{n})$$

Equation (18) represents the interpolation approximation error of *A* with respect to the subspace *X_n_*.
(19)$$d_{A}(x)\equiv \underset{X_{n}\subset X}{\inf}G_{A}(X_{n})$$

Equation (19) represents the interpolation approximation error of *A*. If an *n*-dimensional subspace *X_n_* achieves the infimum of Equation (19), then *X_n_* is referred to as the optimal interpolation approximation subspace of the set *A*, denoted as *X_A_^Φ^*. The basis set {*a_i_*(*M*)}_1_^2^ that achieves the infimum of Equation (18) in *X_A_^Φ^* is called the optimal interpolation approximation basis set of *A*, denoted as {*a_i_*^*^(*M*)}_1_^2^. The following conclusion about the optimal interpolation approximation operator for set *A* holds:(20)$$H_{n}^{A}(M)=\sum_{i=1}^{n}a_{i}^{*}(M)f_{i}$$
where
(21)$$\left\{\begin{array}{l} a_{i}^{*}(M)=\sum_{j=1}^{i}\beta_{ij}R_{j}(M)\;\;i=1,2,\cdots n\\ f_{i}=\sum_{j=1}^{i}\beta_{ij}I_{j}\;\;I_{j}\;is\;determined\;by\;Equation\;\left(15\right).\\ R_{j}(M)=R_{x_{j}}(\xi )R_{y_{j}}(\eta ) \end{array}\right.$$

Here, *β_ij_* is a set of *Schemidt* orthogonalization coefficients, where *i*, *j* = 1, 2,……*n*. *R_j_*(*M*) is the reproducing kernel expression for this bivariate reproducing kernel space.

For any *u*(*M*) ∈ *W*_2_^1^, the following conclusion holds:(22)$$I_{i}(H_{n}^{W}u)=u_{i}\;\;i=1,2,\cdots ,n$$
where *I_i_* is a set of bounded linear functionals defined on *W*_2_^1^.

If {*M_i_*}*_1_^n^* is dense in the interval Ω^2^, then the following conclusion holds:(23)$$\underset{n\to \infty }{\lim}(H_{n}^{U}u)(M)\underline{\underline{u.a.}}\;u(M),\;s.t.,u(M)\in W_{2}^{1},M\in \Omega^{2}$$

Here, “*u.a.*” denotes uniform approximation. Therefore, Equation (20) represents the optimal approximation in *W*_2_^1^ for any given functional over any subspace. The following sections will provide proofs for the above three conclusions:
(1)From Equation (21), it can be seen that {*R_j_*(*M*)}_1_*^n^* is a linearly independent set of functions. By orthogonalizing it using the *Schemidt* method, we obtain a standard orthogonal function system {*a_i_*^Φ^(*M*)}_1_*^n^*, such that:(24)$$a_{i}^{*}(M)=\sum_{j=1}^{i}\beta_{ij}R_{j}(M)\;\;i=1,2,\cdots n,\beta_{ij}\;is\;a\;real\;number$$

Let *X_n_^Φ^* = span[{*a_i_*^*^(*M*)}]*_1_^n^*. Let *ρ* be the upper bound of *u*(*M*). By the *Riesz* representation theorem, we have:(25)$$\begin{array}{ll} d_{A}(M) & =\underset{X_{n}\subset W_{2}^{1}}{\inf}\underset{\begin{array}{l} a_{i}(M)\in X_{n}\\ 1\leq i\leq n \end{array}}{\inf}\underset{u\in U}{\sup}\left|u(M)-\sum_{i=1}^{n}a_{i}(M)I_{i}u\right|\\ & =\underset{X_{n}\subset W_{2}^{1}}{\inf}\underset{\begin{array}{l} a_{i}(M)\in X_{n}\\ 1\leq i\leq n \end{array}}{\inf}\underset{u\in U}{\sup}\left|\left(u(M_{0}),R_{M}(M_{0}))-\sum_{i=1}^{n}a_{i}(M)(u(M_{0}),R_{M_{i}}(M_{0})\right)\right|\\ & =\underset{X_{n}\subset W_{2}^{1}}{\inf}\underset{\begin{array}{l} a_{i}(M)\in X_{n}\\ 1\leq i\leq n \end{array}}{\inf}\underset{\left\|u\right\|\in 1}{\sup}\left|\left(u(M_{0}),R_{M}(M_{0}))-\sum_{i=1}^{n}a_{i}(M)R_{M_{i}}(M_{0}\right)\right|\\ & =\rho \underset{X_{n}\subset W_{2}^{1}}{\inf}\underset{\begin{array}{l} a_{i}(M)\in X_{n}\\ 1\leq i\leq n \end{array}}{\inf}\left|R_{M}(M_{0})-\sum_{i=1}^{n}a_{i}(M)R_{M_{i}}(M_{0})\right| \end{array}$$

Since *X_n_^Φ^* is a closed subspace of *W*_2_^1^, it follows from the above expression that:(26)$$d_{u}(M)=\rho \left|R_{M}(M_{0})-(P_{X_{n}^{\phi }})R_{M_{i}}(M_{0})\right|$$

Here, *P_Xn_^Φ^* is the projection operator onto the subspace *X_n_^Φ^*. *ρ* is the upper bound of *u*(*M*). Thus,
(27)$$\begin{array}{ll} d_{A}(M) & =\rho \left|R_{M}(M_{0})-\sum_{i=1}^{n}\left(R_{M}(M_{0}),R_{M_{i}}(M_{0}\right))R_{M_{i}}(M_{0})\right|\\ & =\rho \left|R_{M}(M_{0})-\sum_{i=1}^{n}\left(R_{M}(M_{0}),\sum_{j=1}^{i}\beta_{ij}R_{j}(M_{0}\right))R_{M_{i}}(M_{0})\right|\\ & =\rho \left|R_{M}(M_{0})-\sum_{i=1}^{n}\left[\sum_{j=1}^{i}\beta_{ij}R_{j}(R_{M}(M_{0}),R_{j}(M_{0}\right))]R_{M_{i}}(M_{0})\right|\\ & =\rho \left|R_{M}(M_{0})-\sum_{i=1}^{n}\left[\sum_{j=1}^{i}\beta_{ij}I_{i}R_{M}(M_{0}\right)]R_{M_{i}}(M_{0})\right|\\ & =\rho \left|R_{M}(M_{0})-(\sum_{i=1}^{n}R_{M_{i}}(M_{0})f_{i})R_{M}(M_{0})\right| \end{array}$$

According to Equation (21), we take:(28)$$H_{n}^{A}(M)={\left(P_{X_{n}^{\phi }}\right)}_{M}=\sum_{i=1}^{n}a_{i}^{*}(M)f_{i}$$

This represents the optimal approximation operator for *A*, thus proving Equation (20).
(2)For any *u* **∈** *W_2_^1^*, noting that *a_i_* (*M*) ∈ *X_n_^Φ^*, we have:(29)$$\begin{array}{l} I_{i}(H_{n}^{W}u)=(I_{i}(H_{n}^{W}u),a_{i}(M))=((P_{X_{n}^{\phi }}u)(M),a_{i}(M))\\ =(u(M),(P_{X_{n}^{\phi }}a_{i}(M))(M),)=(u(M),a_{i}(M))=I_{i}u\equiv u_{i}\;i=1,2,\cdots n \end{array}$$

This proves Equation (22).(3)Let {*X_i_*}_1_^∞^ be dense in [*a*, *b*]. From the previous proofs, the standard orthogonal function system in {*a_i_*^*^(*M*)}_1_*^∞^*, is as follows:(30)$$a_{i}^{*}(M)=\sum_{j=1}^{i}\beta_{ij}R_{j}(M)\;\;i=1,2,\cdots n$$
where *R_j_*(*M*) **∈** *W*_2_^1^. For *u* **∈** *W*_2_^1^, the following holds:(31)$$(u(M),R_{j}(M))=I_{j}u=u(M_{j})\;j=1,2,\cdots$$

Thus, {*a_i_*^*^(*M*)}*_1_^∞^* is a complete system in *W*_2_^1^. If there exists *u* **∈** *W*_2_^1^ such that:(32)$$(u(M),a_{i}^{*}(M))=0\;\;i=1,2,\cdots$$

Based on Equations (31) and (32), we have:(33)$$\sum_{j=1}^{i}\beta_{j}(u(M),R_{j}(M))=\sum_{j=1}^{i}\beta_{j}I_{j}u=\sum_{j=1}^{i}\beta_{j}u(M_{i})=0\;i=1,2,\cdots$$

Therefore, *u*(*M_i_*) = 0. From the density of {*X_i_*}*_1_*^∞^ in [a, b] and the continuity of *u*(*M*), it follows that *u*(*M*) ≡ 0.

Let *n* be a natural number, and *X_n_^Φ^* = span[{*a_i_*^*^(*M*)}]_1_*^n^*. Then, for any *u* **∈** *W*_2_^1^, we have:(34)$$\begin{array}{l} \left\|u(M)-(H_{n}^{A}u)(M)\right\|=\|u(M)-(P_{X_{n}^{\phi }}u)(M)\|\\ =\left\|u(M)-\sum_{i=1}^{n}\left(u(M),a_{i}^{*}(M\right))a_{i}^{*}(M)\right\|\to 0\;n\to \infty \end{array}$$

It follows that:(35)$$\underset{n\to \infty }{\lim}(H_{n}^{U}u)(M)\underline{\underline{u.a.}}\;u(M)\;M\in \Omega$$

This proves Equation (23).

This paper will use the definition and optimal interpolation approximation based on the aforementioned bivariate reproducing kernel space to determine the optimal interpolation approximation method for any number of boundary measurement values in the detection field Ω with given measurement values.

## 3. Optimal Interpolation Approximation for ERT Measurements

This chapter first determines the optimal interpolation approximation of any measurement distribution function *u*(*M*) on the boundary ∂Ω of the ERT sensitive field Ω. It then analyzes the application properties and characteristics of this approximation method.

### 3.1. Interpolation Approximation of the Measurement Distribution Function

Mathematically, the ERT imaging process is essentially about finding an analytical solution for *σ* in Ω by solving the differential Equation (1). According to the uniqueness theorem [36], there is a unique solution to (1) when all values of the boundary function *u*(*M*) on the boundary ∂Ω are given. However, in practice, only a finite number of measurement values of *u*(*M*) can be obtained on ∂Ω, making it impossible to find an analytical solution for *σ*. Currently, in practical engineering applications, *σ* can only be solved numerically using a finite number of measurements. Let *F*(*M*) be the antiderivative of *u*(*M*). This paper first derives the numerical expression for *F*(*M*) based on the fundamental properties of reproducing kernels and then determines *u*(*M*) by finding the derivative of *F*(*M*). It is demonstrated that, when the number of boundary measurements is sufficiently large, the approximated *F*(*M*) can uniformly converge to the antiderivative of *u*(*M*). As illustrated in Figure 3.

For a 16-electrode ERT system, consider exciting electrode *T*_1_ and obtaining 15 measurement values from the other electrodes. Let the origin of the Cartesian coordinate system be defined at the origin, with the 16 measurement electrodes corresponding to the Cartesian coordinates *M_i_* = (*x_i_*, *y_i_*), where *i* = 1, 2,…, 16. The standard equation of a circle in Cartesian coordinates is given by:(36)$$x^{2}+y^{2}=r^{2}$$

The center of the circle is at the origin of the Cartesian coordinate system, and *r* is the radius of the circle. To determine the Cartesian coordinates of electrode *T*_1_, denoted as the initial point (*x*_0_, *y*_0_), and to correctly handle the quadrant issues in Cartesian coordinates, the angle can be calculated using the arctangent function:(37)$$\theta_{0}=\arctan2(y_{0},x_{0})$$

The 16 electrodes are uniformly distributed on the sensitive field Ω (a unit circle), with each pair of adjacent electrodes separated by an angular interval of π/8. The polar angle corresponding to each electrode is given by:(38)$$\begin{array}{l} \theta_{i}=\theta_{0}+i\cdot \theta_{step}\;i=1,2,\cdots 15\\ \theta_{step}=\pi /8 \end{array}$$

The rectangular coordinates of each point can be calculated as:(39)$$\left(x_{i},y_{i})=(r\cos(\theta_{i}),r\sin(\theta_{i})\right)$$

Let *M* = (*x*, *y*), then the following relationships hold:(40)$$u_{i}=F(M_{i})\;i=1,2,\cdots 15$$

Since the rectangular coordinates of any point on the boundary ∂Ω satisfy the Equation (36), the position of this point can be uniquely determined by *x* and *y*. Therefore, the boundary measurement values *u* are functions of *M* = (*x*, *y*) and can be expressed as Formula (40), where Ω^2^ ∈ [0, *r*]X[0, *r*]. In the following discussion, the domain Ω is considered as the unit circle, i.e., r = 1. For a 16-electrode system, besides the 15 measurement values *u_i_*, the value of *F*(*M*) concerning the variable *M* is unknown. However, the results in this section demonstrate that it can be optimally approximated within the function space *W*_2_^1^.

Assume that *F*(*M*) belongs to *W*_2_^1^. This is a very general condition, and almost all application functions in the ERT process can meet this condition. Therefore, *F*(*M*) can be approximated using interpolation based on reproducing kernels, and then *u*(*M*) can be determined from *F*(*M*). Assume *F*(*M*) ∈ *W*_2_^1^, and that:(41)$$W_{2}^{1}(\Omega^{2})=\left\{F(x,y)\left|F,\frac{\partial F}{\partial x},\frac{\partial F}{\partial y},\frac{\partial^{2}F}{\partial x\partial y}\in L^{2}(\Omega^{2})\right.\right\},\Omega^{2}=\left[-1,1\right]\times \left[-1,1\right]$$

The inner product ⟨·,·⟩ and norm ∥·∥ are defined as:(42)$${\langle F(M),G(M)\rangle }_{2}=\int_{-1}^{1}\int_{-1}^{1}FGdxdy+\int_{-1}^{1}\int_{-1}^{1}\frac{\partial F}{\partial x}\frac{\partial G}{\partial y}dxdy+\int_{-1}^{1}\int_{-1}^{1}\frac{\partial F}{\partial y}\frac{\partial G}{\partial x}dxdy+\int_{-1}^{1}\int_{-1}^{1}\frac{\partial^{2}F}{\partial x\partial y}\frac{\partial^{2}G}{\partial x\partial y}dxdy$$
(43)$${\left\|\cdot \right\|}_{2}\underline{\underline{def}}{\langle \cdot ,\cdot \rangle }_{2}^{\frac{1}{2}}$$

Thus, *W*_2_^1^ is a Hilbert space with a two-dimensional reproducing kernel, and the reproducing kernel expression is given by:(44)$$R(M,M_{0})=R_{x_{0}}(x)R_{y_{0}}(y),x,y,x_{0},y_{0}\in \left[-1,1\right]$$
(45)$$R_{x_{0}}(x)=\frac{1}{2sh(2)}\left[ch(x+x_{0})+ch(\left|x-x_{0}\right|-2)\right]$$
(46)$$R_{y_{0}}(y)=\frac{1}{2sh(2)}\left[ch(y+y_{0})+ch(\left|y-y_{0}\right|-2)\right]$$

According to the conclusions proven in Section 2.2, given a sample {(*M_i_*, *u_i_*)}_1_*^n^* and a set of actual measurements *u_i_* = *F*(*M_i_*), the optimal numerical interpolation function for *F*(*M_i_*) is defined as:(47)$$F_{n}(M)=\sum_{i=1}^{n}a_{i}(M)u_{i}\;s.t.,a_{i}(M)=\int_{0}^{M}R_{M_{i}}^{⊥}(t)dt$$

The function *F_n_*(*M*) satisfies:(48)$$u_{i}=F_{n}(M_{i})$$

If the set {*M_i_*}*_1_^n^* is dense in the interval [0, *r*]X[0, *r*], then *F_n_*(*M*) converges to *F*(*M*) uniformly.

### 3.2. Analysis of the Optimality of Interpolation Approximation

To validate the optimality of the interpolation method, four comparative experiments were conducted using COMSOL 3.5a software. Compare the interpolation curve obtained using Equation (47) with the actual curve. For clarity, the numerical values obtained from interpolation are referred to as numerical measurements. Taking Figure 4b as an example, the conductivity of the measured object (four small circles) and the background were set to 1 × 10^−5^ and 1 μS/cm, respectively. A single excitation–single measurement 16-electrode ERT system was used for testing. One electrode was arbitrarily selected for excitation, and the remaining 15 electrodes provided 15 measurement values. Using these 15 measurements, interpolation was performed based on Equation (47), ensuring that interpolation points were uniformly distributed across the entire boundary of the sensitive field and also evenly distributed between two electrodes. This resulted in *n* numerical measurements. By iterating over all 16 electrodes, a total of 16*n* numerical measurements and 240 actual measurements were obtained, resulting in a total of 240 + 16*n* measurements. For this group, *n* = 16, the number of measurements in one set is 31 (represented by the red curve in Figure 4a). For comparison, by inserting *m* electrodes between each of the original 16 electrodes in the model constructed in COMSOL, a total of 240 + 256*m* actual measurements can be obtained. When *m* = 1, the number of measurements in one set is also 31 (represented by the blue curve in Figure 4a). When *n* = 32 and *m* = 2, the number of measurements per set was 47. The comparison results are shown in Figure 4b. The measured object was of a “V” shape, and repeating the above steps resulted in the results shown in Figure 4c,d.

As shown in Figure 4, regardless of whether the shape of the measured model is altered or the number of interpolation points is changed, the curves of numerical measurements closely resemble the actual measurement curves. Both sets of curves are nearly concave functions, and their trends are generally consistent. The error between the numerical measurement curves and the actual measurement curves in all four figures is primarily concentrated at the ends of the U-shaped curves. This discrepancy arises because the measurements at these ends are near the excitation electrodes and are therefore subject to some degree of influence. The shape and trend of the difference curves in the four comparative experiments align with the actual values, confirming that the interpolation method based on the bivariate reproducing kernel is effective for obtaining numerical measurements.

Although the proof process of this interpolation function allows for the insertion of any number of numerical measurements along the boundary of the sensitive field, in practical applications, this insertion does not take into account the limitations imposed by the finite size of the measurement electrodes and the limited space of the measurement boundary. Additionally, it fails to consider that increasing the number of measurements inevitably leads to a reduction in the signal-to-noise ratio (SNR) of the ERT system, resulting in lower spatial resolution. Therefore, it is essential to select an effective and feasible number of numerical measurements while considering the SNR and the constraints of the actual measurement space. In an ERT system, the SNR is typically calculated from a set of measurement values, specifically:(49)$$SNR=10\log{\left\{\sum_{l=1}^{L}{\left[C(l)\right]}^{2}/\sum_{l=1}^{L}\left[C(l\right)-\bar{C}\right]}^{2}\}$$

Here, *C*(*l*) represents the mean of the measured values corresponding to the 1st frame of the ERT image for the same detection field, and $\bar{C}$ is the average of the measured values corresponding to all *L* frames of the ERT images. Here, *l* = 1, 2,…, *L*, and *L* is the total number of frames of ERT images used for testing and calculating the signal-to-noise ratio (SNR). This paper uses a 16-electrode ERT system, which can obtain 240 measurement values in a single measurement (16 × 15 = 240). Therefore, one frame refers to the set of measurement values that contains all the information for a reconstructed image, totaling 240 measurement values. According to the design principle of Equation (49), in order to measure the difference between actual measurements and numerical measurements, a relative error indicator is defined as follows:(50)$$ERROR=\sum_{p=1}^{M}\left|u_{p}^{num}-u_{p}^{sim}\right|/u_{0}^{sim}$$
where *u_p_^num^* and *u_p_^sim^* are a pair of original measurements and numerical measurements, *p* = 1, 2,…, *M*, and *M* is the total number of measurements; *u*_0_*^sim^* is the average of all actual measurements.

Figure 5 illustrates the error variation curves for four different models as the number of numerical measurements increases. Analysis of these interpolation curves reveals that each curve exhibits a minimum point, indicating the existence of an optimal number of numerical measurements. Additionally, as the number of measurements increases, the error gradually decreases. However, due to differences in the distribution of the measured objects and the inherent errors of the imaging algorithms, it is only possible to determine that the optimal number of numerical measurements for the four distributions is around 496. Furthermore, it is observed that as the number of measurements increases, the error variation will also gradually diminish, approaching a stable value.

Given the variation in signal-to-noise ratio (SNR) and error with changes in the number of numerical measurements, it is imperative to carefully select the optimal number of numerical measurements. Conversely, once the number of numerical measurements is determined, it is necessary to subdivide the domain Ω into smaller units to enhance the spatial resolution of ERT reconstruction. Accordingly, the sensitivity matrix must be recalculated to account for the increased number of measurements and pixels in Ω. In this study, the total number of pixels in Ω is set to three times the total number of measurements to investigate the relationship between the increase in the number of measurements and the spatial resolution of ERT.

In the ERT imaging process, the implementation steps for the optimal interpolation approximation method based on the bivariate reproducing kernel are as follows:
**Input:** A set of actual measurements in the sensitive field Ω: {(*x_i_*, *y_i_*), *u_i_*}**Output:** Reconstructed conductivities of all cells in Ω: {*σ_j_*}***Step 1*:** Obtain a set of actual measurements in the sensitive field Ω;***Step 2*:** Determine the number of inserted numerical measurements and their positional coordinates relative to the original electrodes in the sensitive field Ω;***Step 3*:** Perform interpolation using Equation (47) to obtain a set of numerical measurements;***Step 4*:** Partition the detection field Ω into cells, with the total number of cells being three times the total number of measurements.***Step 5*:** Compute the sensitivity matrix *S* corresponding to the number of measurements;***Step 6*:** Reconstruct the conductivities of all cells in Ω using the selected ERT algorithm;***Step 7*:** Validate the algorithm by error assessment according to Equation (50).

## 4. Results

In this section, the image reconstruction quality of the BRKBI method is evaluated through both simulation and real model measurement. The evaluation method involves comparing the relative error in spatial resolution between the original image and the reconstructed image. The error evaluation metric, RE, is defined as follows:(51)$$RE=\sum_{i=1}^{n}\left|\sigma_{i}-\sigma_{i}^{*}\right|/\sigma_{i}^{*}\;i=1,2,\cdots ,n$$

In Equation (51), ***σ_i_*** represents the electrical conductivity of all pixels within Ω obtained through the reconstruction algorithm, while ***σ_i_**** represents the actual electrical conductivity of the corresponding pixels in the simulation or experiment. A smaller RE value indicates better reconstruction quality. In the current ERT research field, RE is widely used to evaluate imaging quality.

### 4.1. Simulation

Using COMSOL software, a 16-electrode ERT system was constructed, and seven models with different distribution characteristics were created within the measurement field Ω. In all models, the conductivity of the targets and the background were set to 1 × 10^−5^ and 1 μS/cm, respectively. The 240 measurements generated by the 16-electrode system were treated as the actual measurements. The RKBI and BRKBI methods were applied to interpolate additional measurement values. The interpolation procedure involved inserting one to three measurement points between each pair of electrodes in the 16-electrode ERT system, resulting in 496, 752, and 1008 interpolated values, respectively. These interpolated values, combined with the actual measurements, were used to reconstruct the images, and the image reconstruction quality was assessed. To verify the general applicability of the BRKBI method, the reconstruction algorithms selected for imaging were the LBP and Tikhonov methods. To ensure the objectivity and accuracy of the experimental results, all models using the Tikhonov algorithm in the experiments are adjusted such that the RE values represent the optimal image reconstruction quality for the current number of measurements and imaging parameters.

As shown in Table 1, Models M1–M3 represent discrete models, while Models M4–M7 represent continuous models. In the images, the red regions correspond to the targets, and the blue regions correspond to the background. The first row of Table 1 presents the imaging results obtained using the LBP algorithm with 240 actual measurements from 16 electrodes. The subsequent rows show the imaging results based on 496, 752, and 1008 interpolated measurements using the RKBI and BRKBI methods, respectively, with the LBP algorithm applied. Each imaging result is accompanied by its corresponding RE value. For the discrete models M1–M3, both the BRKBI and RKBI methods accurately display the position and shape of the models. When the number of interpolated measurements reaches a certain level, BRKBI can distinguish the number of small circles within the model more effectively than RKBI and generally produces clearer images. For the continuous models M4–M7, both BRKBI and RKBI show the approximate contours and locations of the targets. However, many of the imaging results for both the discrete and continuous models are not sufficiently clear to accurately determine the number and edges of the targets. This lack of clarity is primarily due to the low spatial resolution of the LBP imaging algorithm. However, evaluating the imaging quality solely based on visual inspection is not entirely objective; it is also essential to analyze the RE values associated with each image.

Analyzing the RE values of the imaging results in Table 1 reveals the following: For all seven models, the RE values obtained using the BRKBI method are consistently lower than those derived from the actual measurements in the first row. When comparing the RE values between BRKBI and RKBI, 18 out of 21 imaging results show that the RE values from BRKBI are lower than those from RKBI, which demonstrates the effectiveness of the BRKBI method. As the number of interpolated measurements increases, the imaging results for certain models exhibit some improvement. However, the quality of the reconstructed images does not continuously improve with the increase in measurement numbers. This phenomenon is explained by the signal-to-noise ratio (SNR) discussed in Section 3.2. Among the seven models, the RE values for models M1, M4, M5, M6, and M7 reach their minimum around 496 interpolated measurements, while the RE values for models M2 and M3 reach their minimum around 752 interpolated measurements.

Table 2 presents the imaging results using the Tikhonov algorithm, structured similarly to Table 1. The first row in Table 2 displays the imaging results using the Tikhonov algorithm with 240 actual measurements obtained from 16 electrodes. Subsequent rows show imaging results based on 496, 752, and 1008 interpolated measurements using both the RKBI and BRKBI methods. Each imaging result is accompanied by its corresponding RE value.

Comparing the imaging results in Table 2 with those in Table 1, it is evident that the imaging quality of all seven models in Table 2 is superior. This improvement is attributed to the superior imaging performance of the Tikhonov algorithm compared to the LBP algorithm. For the discrete models M1–M3, both BRKBI and RKBI can reveal the number, position, and shape of the models, but the reconstructed images using BRKBI exhibit sharper edges and fewer artifacts. For the continuous models M4–M7, the imaging quality after interpolation using BRKBI and RKBI is lower than that of the discrete models. However, for the same number of measurements, the images reconstructed using the BRKBI method more closely resemble the original images in terms of shape.

An analysis of the RE values in Table 2 shows that the BRKBI method outperforms the RKBI method in 18 out of the 21 imaging results, further demonstrating the effectiveness of the BRKBI method. Comparing the models’ positions, shapes, edge characteristics, and RE values in the images, the Tikhonov algorithm used in Table 2 results in overall lower RE values for the seven models than when using the LBP algorithm, as seen in Table 1. Specifically, the RE values for models M1, M2, M3, and M6 reach their lowest around 1008 measurements, for M7 around 752 measurements, and for M4 and M5 around 496 measurements.

### 4.2. Real Model Measurements

The real model measurements were conducted using a 16-electrode ERT measurement system developed in the laboratory. This system consists of a PC for data processing and image reconstruction, an FPGA-based data acquisition system, and a container equipped with 16 excitation electrodes. Each electrode has a diameter of 10 mm, and the cylindrical container has a diameter of 320 mm (as shown in Figure 6).

The real model measurements were divided into four groups: the first two groups, M1 and M2, represent discrete models, while the latter two groups, M3 and M4, represent continuous models. These models were fabricated using UV-curable resin material via 3D printing (as shown in the first row of Table 3) to simulate objects with a conductivity close to 0 μS/cm. During the experiments, the container was filled with saline solution with a conductivity of 0.01 μS/cm. Given the superior imaging performance of the Tikhonov regularization, the Tikhonov algorithm was used for all reconstructions in this study. The second row of Table 3 presents the imaging results obtained using the Tikhonov algorithm from 240 actual measurements collected by the ERT system. Subsequently, RKBI and BRKBI methods were used to generate 496, 752, and 1008 measurements, which were then used for imaging with the Tikhonov algorithm. To quantitatively evaluate imaging quality, this experiment uses a scanned copy of the real model to create a model in COMSOL software, which is used to calculate the RE values of the reconstructed images. Each imaging result is accompanied by its corresponding RE value. The following analysis discusses the imaging quality for the four models.

➀Model 1

Analysis of the RE values indicates that, with an increase in the number of numerical measurements, the RE value for BRKBI initially decreases and then increases. The RE values obtained from the three sets of interpolated measurements are consistently lower than the RE value from the original 240 measurements. In comparison to RKBI, the RE values for BRKBI are consistently lower under the same numerical measurement conditions, demonstrating the advantages of the BRKBI method. Examining the image reconstruction results, it is evident that the BRKBI images exhibit minimal artifacts, with relatively clear image contours and positions that are consistent with the original image. In contrast, the RKBI images with 496 measurements display artifacts and lack clarity. Although the image quality improves with an increased number of numerical measurements, it does not reach the accuracy and stability of the BRKBI results.

➁Model 2

Analysis of the RE values indicates that, as the number of numerical measurements increases, the RE value for BRKBI initially decreases and then increases. For the three sets of interpolated measurements, the RE values are lower than those obtained from the original 240 measurements, except for the set with 1008 measurements. Compared to RKBI, the RE values for BRKBI are lower under all conditions except for the 1008 measurement set, highlighting the superiority of the BRKBI method. Regarding the image reconstruction results, both BRKBI and RKBI methods exhibit artifacts in the reconstructed images, and the overall quality is less than ideal. However, the BRKBI method is capable of distinguishing the distribution and number of objects in the images. In contrast, the RKBI method produces irregular shapes with 496 measurements, and with 752 and 1008 measurements, it results in ring-like shapes. All three sets of RKBI images differ significantly from the original images. Therefore, BRKBI demonstrates better performance than RKBI for this model.

➂Model 3

The analysis of the RE values indicates that the RE values for the three sets of interpolated imaging using BRKBI are significantly lower compared to the RE value for imaging with the original 240 measurements. In comparison to RKBI, BRKBI exhibits lower RE values under all conditions except for the set with 752 measurements, demonstrating the advantages of the BRKBI method. In terms of image reconstruction, the three sets of images reconstructed using BRKBI accurately reflect the position and shape of the original image. In contrast, the images reconstructed with RKBI only reveal the central crossing point of a “plus” shape, with the four edges not being visible. Consequently, BRKBI provides superior results compared to RKBI for this model.

➃Model 4

The analysis of RE values indicates that the RE values for the three sets of interpolated imaging using BRKBI show a significant decrease compared to the RE value for imaging with the original 240 measurements. When compared to RKBI, BRKBI consistently yields lower RE values across all measurement conditions, highlighting the advantages of the BRKBI method. In terms of image reconstruction, the three sets of images reconstructed with BRKBI are consistent with the original image’s position, though the lower edge of the object remains somewhat unclear. Compared to the RKBI method, the BRKBI-reconstructed images have clearer contours. Thus, for this model, BRKBI outperforms RKBI.

Overall, the quality of images reconstructed using interpolation methods is superior to that obtained from the original 240 measurements. Between the two interpolation methods, BRKBI reconstructed images are, in general, superior to the RKBI images in terms of position, shape, edges, and RE values.

**Table 3 sensors-24-07165-t003:** Reconstructed image with real data using Tikhonov, RKBI, and BRKBI algorithms.

	M1–4	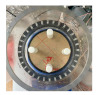	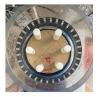	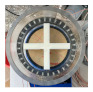	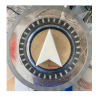
Totalof Number Measurement	
240	Tihonov	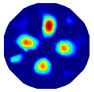	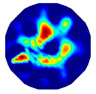	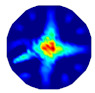	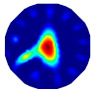
RE	0.2178	0.2339	0.2276	0.2237
496	RKBI	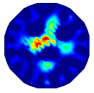	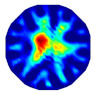	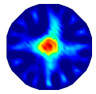	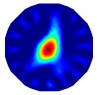
RE	0.2281	0.2327	0.2154	0.2371
BRKBI	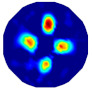	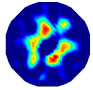	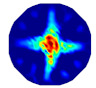	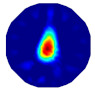
RE	0.2120	0.2208	0.2061	0.2149
752	RKBI	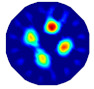	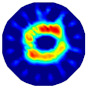	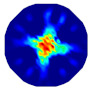	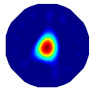
RE	0.2081	0.2337	0.1991	0.2403
BRKBI	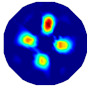	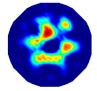	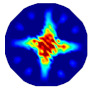	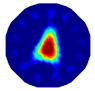
RE	0.2076	0.2193	0.2134	0.2178
1008	RKBI	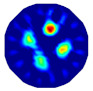	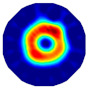	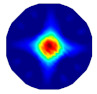	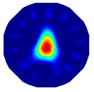
RE	0.2176	0.2386	0.2162	0.3484
BRKBI	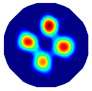	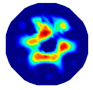	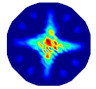	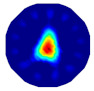
RE	0.2105	0.2456	0.2076	0.2109

## 5. Discussion

Based on the limitations of existing univariate radial kernel interpolation functions, such as limited approximation capability and low reconstruction image resolution, this paper proposes a bivariate radial kernel interpolation method. The theoretical derivation of the interpolation model and error bounds is provided, and the optimality of this method is proven. The effectiveness of the BRKBI (Bivariate Reproducing Kernel-Based Best Interpolation) method is verified through comparisons of reconstructed images in both simulation and actual measurement platforms, evaluating characteristics such as position, shape, and RE values. The key findings from the experiments are summarized as follows:Selection of Imaging Algorithms: The choice of imaging algorithm is crucial for image reconstruction results. The same interpolation method yields different final image quality depending on the imaging algorithm used.Optimal Number of Measurements: The optimal number of numerical measurements depends on the model type and the imaging algorithm. It is necessary to consider the model type and the imaging algorithm when determining the best number of measurements.

The proposed interpolation algorithm further reduces approximation errors and improves image resolution. However, when dealing with a large number of images or complex shapes, the quality of image reconstruction still requires improvement. Therefore, enhancing existing imaging algorithms to improve image quality remains a direction for future research.

## Figures and Tables

**Figure 1 sensors-24-07165-f001:**
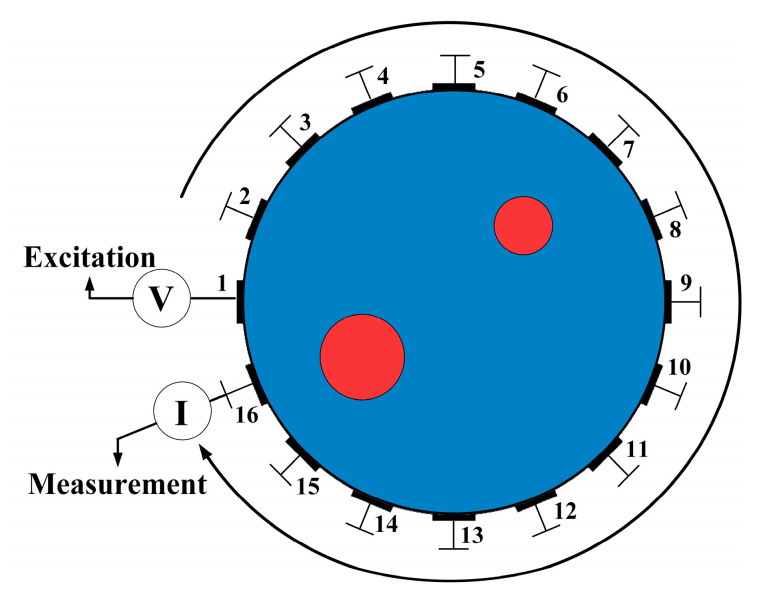
Principle Diagram of a 16-Electrode ERT System.

**Figure 2 sensors-24-07165-f002:**
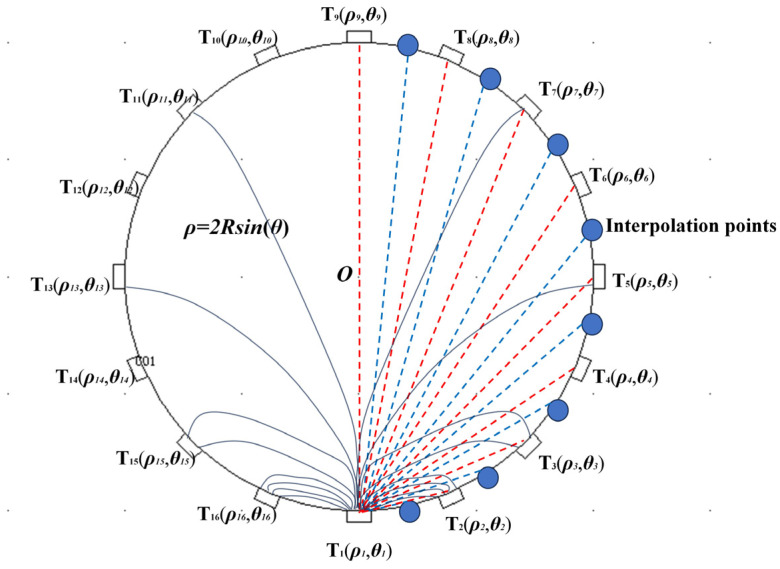
Explanation of the RKBI Method.

**Figure 3 sensors-24-07165-f003:**
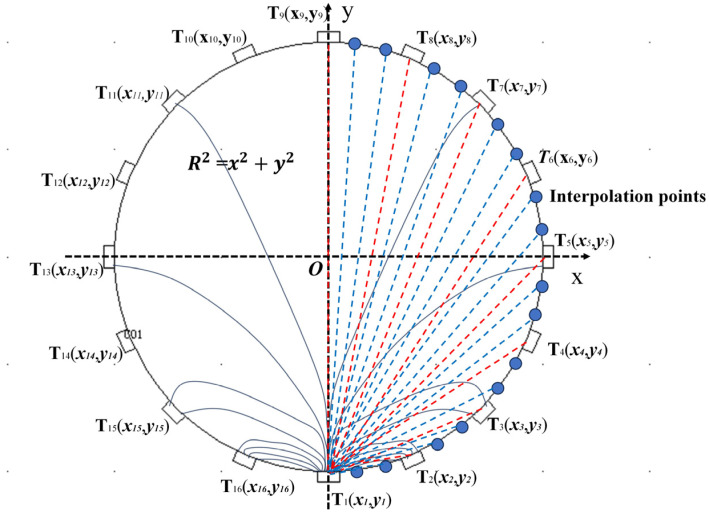
Explanation of the BRKBI Method.

**Figure 4 sensors-24-07165-f004:**
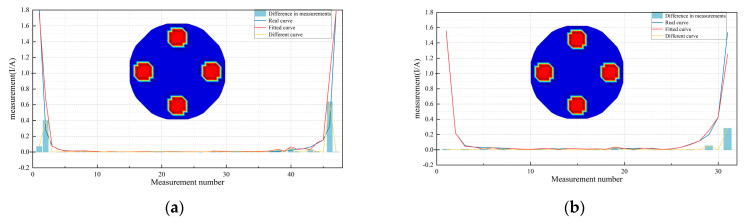
Comparison of numerical and simulated measurement along the two models. (**a**,**b**) 4−circle model. (**c**,**d**) V−shaped model.

**Figure 5 sensors-24-07165-f005:**
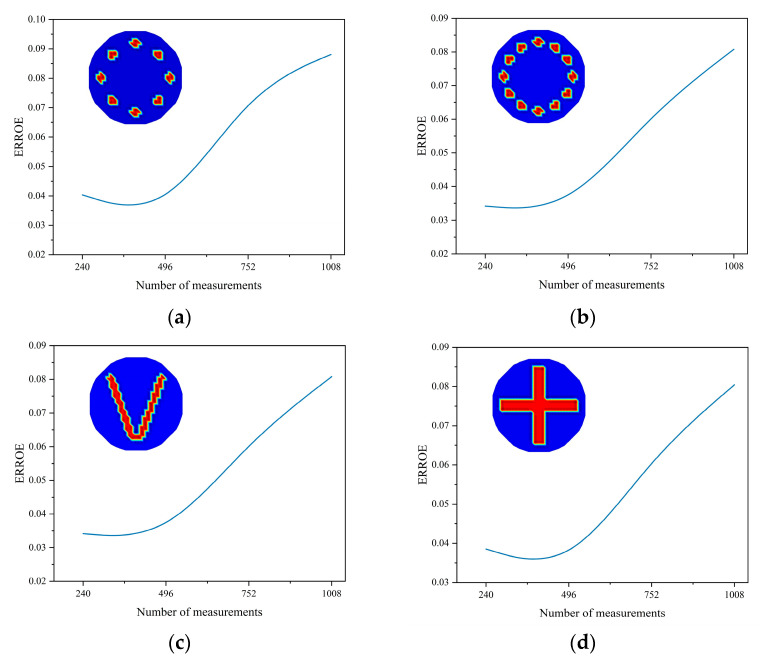
*Error* of various numerical measurements. (**a**) 8-circle model. (**b**) 12-circle model. (**c**) V-shaped model. (**d**) Cross-shaped model.

**Figure 6 sensors-24-07165-f006:**
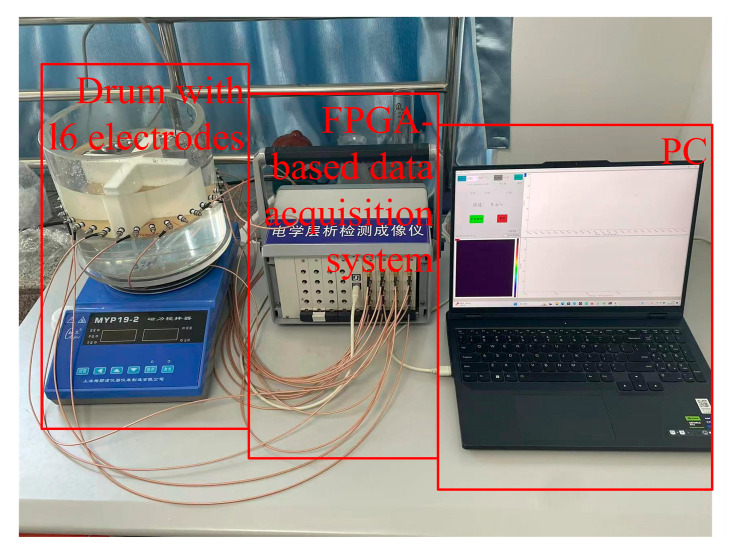
Sixteen-electrode ERT experiment system.

**Table 1 sensors-24-07165-t001:** Reconstructed image with artificial data using LBP, RBKI, and TRBKI algorithms.

	M1–7	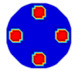	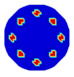	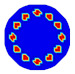	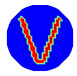	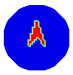	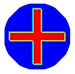	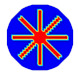
Totalof NumberMeasurement	
240	LBP	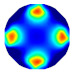	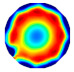	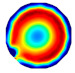	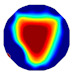	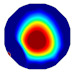	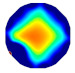	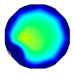
RE	0.2951	0.1865	0.1960	0.2577	0.2600	0.2876	0.2683
496	RKBI	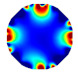	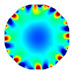	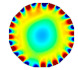	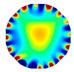	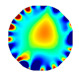	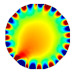	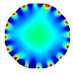
RE	0.3072	0.2451	0.2371	0.1607	0.1862	0.2814	0.2322
BRKBI	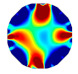	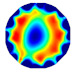	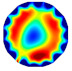	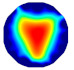	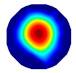	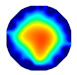	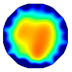
RE	0.2530	0.1518	0.1856	0.1764	0.1144	0.2270	0.2055
752	RKBI	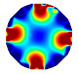	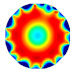	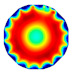	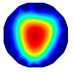	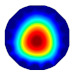	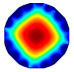	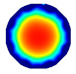
RE	0.2755	0.1379	0.1496	0.3179	0.3621	0.3272	0.2560
BRKBI	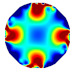	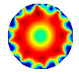	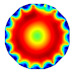	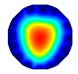	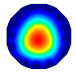	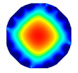	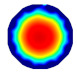
RE	0.2749	0.1310	0.1412	0.2144	0.2499	0.3227	0.2635
1008	RKBI	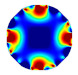	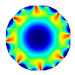	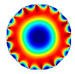	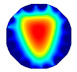	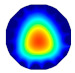	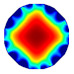	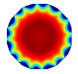
RE	0.3520	0.2263	0.2339	0.2844	0.3616	0.3511	02367
BRKBI	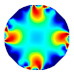	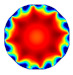	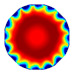	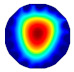	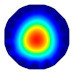	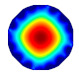	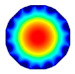
RE	0.3059	0.1868	0.1771	0.2322	0.2898	0.3332	0.2847

**Table 2 sensors-24-07165-t002:** Reconstructed image with artificial data using Tikhonov, RBKI, and TRBKI algorithms.

	M1–7	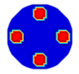	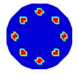	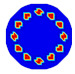	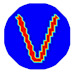	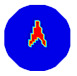	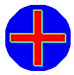	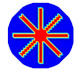
Totalof Number Measurement	
240	Tihonov	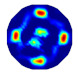	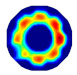	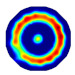	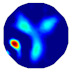	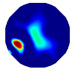	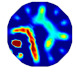	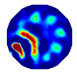
RE	0.3291	0.2176	0.3065	0.2637	0.2580	0.3069	0.4164
496	RKBI	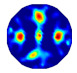	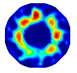	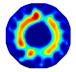	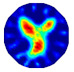	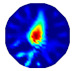	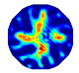	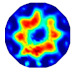
RE	0.3177	0.2529	0.2480	0.3224	0.2889	0.3565	0.2936
BRKBI	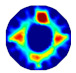	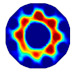	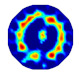	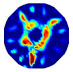	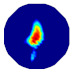	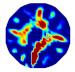	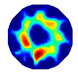
RE	0.2386	0.1645	0.2773	0.2558	0.1070	0.2852	0.2906
752	RKBI	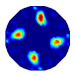	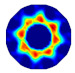	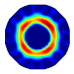	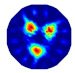	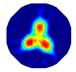	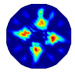	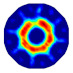
RE	0.2865	0.1661	0.3554	0.3391	0.4230	0.3122	0.2882
BRKBI	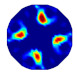	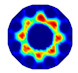	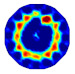	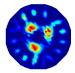	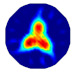	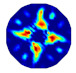	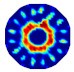
RE	0.2631	0.1764	0.2879	0.3033	0.2026	0.3466	0.2303
1008	RKBI	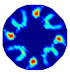	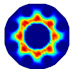	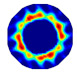	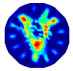	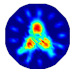	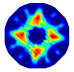	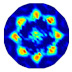
RE	0.2293	0.1755	0.2303	0.3496	0.4651	0.2809	0.4251
BRKBI	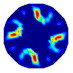	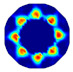	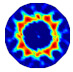	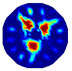	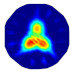	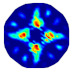	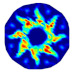
RE	0.2102	0.1632	0.2598	0.3114	0.4291	0.2678	0.2511

## Data Availability

The data presented in this study are available on request from the corresponding author due to the data being obtained through experiments and not being publicly available.

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
