# Peer review of "Application of Bivariate Reproducing Kernel-Based Best Interpolation Method in Electrical Tomography"

_sensors, 2024, doi:10.3390/s24227165_

Round 1
Reviewer 1 Report
Comments and Suggestions for Authors
The authors have proposed a Bivariate Reproducing Kernel-Based Best Interpolation (BRKBI) method to improve the image reconstruction which aims to improve the quality of the reconstructed images. I recommend this work for publications on Sensors after the authors addressed my concerns below:
1. In the introduction, the references did not include the paper published in these two years, especially for the reconstruction algorithms.
2. In the section 2, I think that a lot of contents are not useful. I suggested that authors should focus on the relationship between the influences of measurement number on image quality.
3. In section 2.2, the description of interpolation approximation based on Bivariate Reproducing Kernels is very difficult to relate to obtain more measurement number, and how to prove the accuracy?
4. For the validation by simulation and experiments, it is found that the optimal number of Interpolation will determined by the distribution of tested phantoms. Therefore, how to determine the optimal number of Interpolation before conducting proposed methods.
5. In figure 1, ERT systems usually employ the current excitation and voltage measurement. The excitation and measurement principle of this paper was proposed by Jiangtao Sun( the university of Manchester), please add related reference.
6. Figure 4 and figure 5 are not clear, please update more clear figures
7. The symbols of some formulas did not provide explanations. Such as formula -8, 9,10...35.
Comments on the Quality of English Languageno.
Reviewer 2 Report
Comments and Suggestions for Authors
See attached docx file

Author Response
Please see the attachment. The line numbers referenced in the response document correspond to those in your annotated PDF.

Reviewer 3 Report
Comments and Suggestions for Authors
please see the attched file

Minor editing of English language required.
Round 2
Reviewer 1 Report
Comments and Suggestions for Authors
Authors shoud check my comments more carefully, I think the revised manuscript is too simple, which is not strong enough to defend my comments. please making the revisions again.
Reviewer 3 Report
Comments and Suggestions for Authors
All my issuses was revised by the author, now l think that the paper can be published in Sensors
Comments on the Quality of English LanguageMinor editing of English language required.
Author Response
Dear Reviewer,
Thank you for your recognition of our research and for your positive comments regarding the agreement to publish our manuscript. We sincerely appreciate your invaluable feedback and support.
Sincerely,
Junqi Yu